# Epidemiology of Heart Disease of Uncertain Etiology: A Population Study and Review of the Problem

**DOI:** 10.3390/medicina55100687

**Published:** 2019-10-14

**Authors:** Alessandro Menotti, Paolo Emilio Puddu

**Affiliations:** 1Association for Cardiac Research, 00161 Rome, Italy; amenotti2@gmail.com; 2Department of Cardiovascular, Respiratory, Nephrological, Anesthesiological and Geriatric Sciences, Sapienza University of Rome, 00161 Rome, Italy; 3Equipe d’Accueil (EA) 4650, 14000 Caen, France

**Keywords:** coronary heart disease, heart diseases of uncertain etiology, risk factors, mortality serum cholesterol

## Abstract

*Background and objectives:* Previous epidemiological studies have identified a group of heart diseases (here called heart diseases of uncertain etiology—HDUE) whose characteristics were rather different from cases classified as coronary heart disease (CHD), but frequently confused with them. This analysis had the purpose of adding further evidence on this issue based on a large population study. *Materials and Methods:* Forty-five Italian population samples for a total of 25,272 men and 21,895 women, free from cardiovascular diseases, were examined with measurement of some risk factors. During follow-up, CHD deaths were those manifested as myocardial infarction, other acute ischemic attacks, and sudden death of probable coronary origin, after reasonable exclusion of other causes. Cases of HDUE were those manifested only as heart failure, chronic arrhythmia, and blocks in the absence of typical coronary syndromes. Cox proportional hazards models were computed separately for CHD and HDUE, with 11 risk factors as possible predictors. *Results:* During an average of 7.4 years (extremes 1–16) there were 223 CHD and 150 HDUE fatal events. Male sex, age, smoking habits, systolic blood pressure, serum cholesterol, and plasma glucose were significantly and directly related to CHD events, while high density lipoprotein (HDL) cholesterol was so in an inverse way. The same risk factors were predictive of HDUE events except serum cholesterol and HDL cholesterol. Multivariable hazards ratio of serum cholesterol (delta = 1 mmol/L) was higher in the CHD model (1.24, 95% CI 1.11–1.39) than in the HDUE model (1.03, 0.5% C.I. 0.89–1.19) and the difference between the respective coefficients was statistically significant (*p* = 0.0444). Age at death was not different between the two end-points. *Conclusions:* CHD and HDUE are probably two different morbid conditions, only the first one is likely bound to gross atherosclerotic lesions of coronary arteries and linked to blood lipid levels. We reviewed the problem in epidemiological investigations and addressed inflammation as a potential cofactor to differentiate between CHD and HDUE.

## 1. Introduction

Classification and definition of cardiovascular diseases in epidemiological studies is not univocal. Heart diseases are usually kept alone and segregated from other major conditions, such as stroke and peripheral artery diseases. However, it is not always clear what heart diseases mean, and while they should at least be subdivided into coronary heart disease (CHD) and other heart diseases, a distinction is not firmly stated in several investigations. Moreover, the larger category of cardiovascular diseases (CVD) frequently includes all possible diseases of the heart and circulatory system, independently from respective etiologies. Usually CHD is defined by WHO-ICD-9 codes 410–414 [1], but the meaning of codes called “other forms of chronic ischemic heart disease” is somewhat confusing, or at least not explicit, although it represents an improvement of code 412 (chronic ischemic heart disease) and code 414 (asymptomatic ischemic heart disease) of the previous ICD-8 classification [2].

Nowadays, the WHO ICD-10 classification [3] is universally used in clinical and scientific practice and a better description can be found of the several components of ischemic heart disease, but, again, several components of code I25 (chronic ischemic heart disease) are still open to misinterpretations.

In a series of analyses conducted by our research group on different population studies [4,5,6,7,8,9,10,11,12], we proposed the term and concept of heart disease of uncertain etiology (HDUE) (initially also called atypical CHD), a condition characterized by late occurrence in life, manifested only as heart failure, chronic arrhythmia, and blocks, in the absence of the typical syndromic presentations of CHD (such as angina pectoris, myocardial infarction, other acute ischemic syndromes, or sudden death), not being predicted by serum cholesterol levels, nor by dietary patterns. All of these factors suggested that we are probably facing two etiologically different conditions, that is CHD and HDUE. [4,5,6,7,8,9,10,11,12].

The possible etiology of HDUE is far from being ascertained, known, or understood, and likely is not homogeneous, but identifying the risk factors of the two conditions (CHD and HDUE) might represent a first step in the search of the real etiology. It was of interest to search for further confirmation of this concept by testing this principle on other population studies, that could offer, in front of a larger both-gender overall size, similar types of measured data and end-points, but with quite shorter follow-up, still focusing on specific outcomes, that is mortality due to CHD vs. HDUE.

Our analysis was based on data collected about 30–35 years ago, when the possible role of inflammation CHD and heart failure was not yet considered, and therefore markers of inflammation were not measured. However, we decided to stress the existence of cases we call HDUE, that are poorly classified and considered and also need the attention of research from the point of view of the role of inflammation in their etiology and/or pathogenesis. We, therefore, also take this opportunity to review the problem with a special focus on HDUE and the potential differentiation that inflammatory markers may add vs. CHD.

## 2. Materials and Methods

The data used here derive from the Italian studies contributing to the Euro-SCORE project [13] that were performed within the RIFLE project (risk factors and life expectancy) [14].

The Euro-SCORE was a project sponsored by the European community with the purpose to produce European risk functions for fatal cardiovascular disease using population data from 12 European countries. The RIFLE project was an Italian pooling project combining together 9 population studies, including 52 population samples spread all over the country, whose aim was to study the relationship of many risk factors with fatal cardiovascular diseases and other conditions. Its contribution to the Euro-SCORE was, numerically, the largest among those of the 12 countries, providing 26% of all subjects involved.

Originally the RIFLE project comprised 52 cohorts for a total of 72,549 men and women aged 19–99 years. For the purpose of this analysis, one cohort was excluded because it was withdrawn by the responsible investigators and 6 cohorts were excluded because of incomplete follow-up data. Then, only subjects aged 35–74 years were selected (N = 50,462), but 1738 were excluded because of the documented presence of wrong or largely incomplete data. In the remaining 48,724 subjects, who had 9 personal characteristics measured at entry examination, imputation of some missing data was made using a multivariate normal approach for a total of 9% of data. Subjects aged >74 years were few and most of them belonged to the excluded cohorts. For the purpose of this analysis, we excluded also 1557 subjects carrying a major cardiovascular diseases at entry examination, reaching a total of 25,272 men and 21,895 women (grand total 47,167).

Entry examinations were carried out between 1978 and 1988 and the following risk factors were measured and considered for this analysis: (a) sex: 0 = female, 1 = male; (b) age in years; (c) smoking habits, assessed by a questionnaire derived from the WHO Cardiovascular Survey Methods Manual (WHO Manual) [15]; subjects were classified as dummy variables in three classes, i.e., never smokers (used as refence in multivariate models), ex-smokers, and current smokers; (d) body mass index (BMI), in kg/m^2^, derived from weight and height, measured following the rules of the WHO Manual [15]; (e) systolic blood pressure, in mmHg, measured in sitting position after 5 min rest, on the right arm, using a mercury sphygmomanometer following the procedure described in the WHO Manual [15]; (f) serum cholesterol, in mmol/L, measured on blood sample taken after 12 h fasting—several automated enzymatic methods were employed in the various cohorts, but all of them were under direct or indirect quality control from the WHO Lipid Reference Center in Prague [14]; (g) high density lipoprotein (HDL) cholesterol, in mmol/L, measured using the same methods as for total cholesterol, after precipitation with phosphotungstic acid or with heparin or dextran sulphate in the various cohorts—quality control was guaranteed by the same center as for total cholesterol [14]; (h) serum triglycerides, in mg/dL, measured on fasting sera, using several automated enzymatic methods—the same comments apply as for total cholesterol [14]; (i) plasma glucose, in mg/dL—measures on fasting plasma were measured by several automated methods, the most common being those of Trinder, God-Pap, and God-Perid, a central Italian laboratory supervised for quality control and standardization [14].

Follow-up for life status and mortality was run for an average of 7.4 years (extremes 1–16 years) and causes of deaths were coded using the 9th Revision of the WHO-ICD [1] following defined rules.

ICD codes were assigned by a single coder transforming diagnoses expressed in words into ICD codes. Moreover, in the presence of multiple causes of death and uncertainties about the primary cause, a rank criterium was adopted with violent causes, cancer, CHD, stroke, and others in sequence.

Coronary heart disease (CHD) was defined by ICD-9 [1] codes 410–413, 798.2 corresponding to cases with myocardial infarction, other acute ischemic attacks, angina pectoris, and sudden death of probable coronary origin, after reasonable exclusion other possible causes; code 414—chronic or other forms of CHD—was excluded for reasons given elsewhere [6]. Heart disease of uncertain etiology was defined by ICD-9 [1] codes 402–404, 414, 426–428, corresponding to hypertensive disease (except essential hypertension), chronic or other forms of CHD, conduction disorders, cardiac dysrhythmias, and heart failure—that is, a mix of heart diseases that do not have a defined etiology but frequently are confused with typical CHD.

Written or verbal consensus, depending on different cohorts, was obtained from participants to comply with the principles of the Helsinki Declaration. The RIFLE research group was not controlled by an ethical committee, a procedure that was not used at the time of data collection, but it was under control of the Italian CNR (National Research Council), which was a major sponsor of the study. Moreover, within the RIFLE Research Group, detailed rules were adopted and applied for the use of data, their analysis, and publication.

Data of risk factors and death rates were presented as proportions and standard errors for categorial variables, and as medians and interquartile ranges for continuous variables since they did not have a normal distribution. Comparisons of data between men and women were performed by the test of proportions for categorial data, and by both the *t*-test (on means and standard deviations) and the nonparametric test of Mann–Whitney for continuous variables. Pairs of risk factor multivariable coefficients of the two end-points were compared using the *t*-test. For all statistical tests, significance was set at *p* < 0.05.

Limiting the calculations to the male group (N = 25,272), baseline survival and risk factor coefficients of the Cox models were applied back to the single individuals for the estimate of their risk probabilities for CHD and HDUE. These estimates were used to compute the relative risk across tertiles of estimated risk. Moreover, ROC curves were computed, again only in men, for both end-points.

## 3. Results

Baseline mean values and proportions of the measured risk factors and mortality rates are reported in Table 1, separately for men, women, and their pool, after discarding the prevalent cases, namely those carrying any major cardiovascular disease. Following the Mann–Whitney test and the *t*-test for continuous variables, and the test of proportions for categorical variables, all levels were significantly higher in men than in women except those for HDL cholesterol, where the relationship was reverse. Mortality rates, too, were significantly higher for men than women.

During an average follow-up of 7.4 years (corresponding to an exposure of 348,579 person/years), there were 1647 deaths from all causes (34.9 per 1000), of which 223 were attributed to CHD (4.7 per 1000) and 150 to HDUE (3.2 per 1000). Age at death was 60.7 years for all causes, 61.0 years for CHD events, 61.2 years for HDUE events; the difference between age at death of the two cardiovascular end-points did not reach statistical significance. Mortality from CHD was 6.4-fold higher in men than in women, that from HDUE 4.5-fold.

Cox proportional hazards models for CHD and HDUE mortality as a function of 10 covariates (plus 1 reference) are given in Table 2. Delta, chosen for computation of hazards ratios, did roughly correspond—for continuous variables—to one standard deviation.

In the CHD model, all risk factors had hazards ratios with confidence intervals not including one (statistically significant), except ex-smokers, body mass index, and triglycerides. HDL cholesterol had an inverse relationship with risk of CHD, while male gender, age, current smokers, systolic blood pressure, serum cholesterol, and plasma glucose had a direct relationship.

In the HDUE model, only male gender, age, current smokers, systolic blood pressure, and plasma glucose were significantly associated with the risk to develop that fatal condition. Instead, both total and HDL serum cholesterol were not associated with deaths and the *p*-values of their coefficients were not significant (0.7382 and 0.9297, respectively).

Cox models were also produced separately for men and women, but in the latter case the models did not converge, likely owing to the small number of cases. Cox models for men are given in Table 3, showing rather similar characteristics to those given in Table 2 for the pooled sexes.

Pairs of multivariable risk factor coefficients of CHD and HDUE models found in the pooled sexes were compared by *t*-test, as reported in Table 4. In no case was the difference of multivariable risk factor coefficients statistically significant when comparing CHD with HDUE, except for total serum cholesterol. The difference for HDL cholesterol between CHD and HDUE was not far from significance (*p* = 0.0802). As a consequence, the major and only difference in the predictive power of this group of risk factors of CHD vs. HDUE remained total serum cholesterol, although the inverse predictive role of HDL, borderline statistically significant in the present investigation (Table 4), should also be considered in conjunction with other characteristics derived from different experiences (Table 5).

In a separate model, the two end-points were pooled together and the size of the multivariable coefficient of total serum cholesterol was 0.1399 vs. 0.2184, as found in the previous CHD model (details not reported). This suggests that there was no additive effect on the magnitude of serum cholesterol coefficient when combining the two conditions.

The relationship of serum cholesterol with the two end-points is depicted in Figure 1, where increasing levels were strongly associated with increasing risk for CHD, whereas the relationship was practically flat for HDUE. A similar graph dealing with HDL cholesterol, showed the inverse relationship with CHD and a flat relationship with HDUE (Figure 2).

Limiting the computation to men, we estimated the individual risk for both CHD and HDUE following the procedure described in Material and Methods. For CHD, the average risk in tertile 3 (upper) of the distribution was 0.0248 (95% C.I. 0.0245–0.0251), while it was 0.0030 (95% C.I. 0.0029–0.0031) in tertile 1, with a ratio between the two of 8.3. For HDUE the average risk in tertile 3 was 0.0146 (95% C.I. 0.0144–0.0147), while it was 0.0020 (95% C.I. 0.0019–0.0021) in tertile 1, with a ratio between the two of 7.3.

The ratio of the number of cases in tertile 3 vs. those in tertile 1 was 9.9 for CHD and 10.0 for HDUE.

The ROC curves derived from the Cox models of men produced levels of 0.73 for CHD and 0.72 for HDUE.

## 4. Discussion

### 4.1. Confirmatory Finding

This analysis was run to provide confirmation of previous findings obtained in different population studies, roughly run during the same time limits, that is the mid part of the second half of last century [4,5,6,7,8,9,10,11,12]. Nowadays the situation of morbidity and mortality and the standards of treatment are surely different, but it is unlikely that the “etiology” of these two groups of heart diseases has substantially changed together with their determinants.

Findings of this analysis have shown that CHD and HDUE have different associations with predictive risk factors, specifically total serum cholesterol and HDL cholesterol, while the predictive roles of blood pressure, smoking habits, and plasma glucose were not significantly different. These were the main findings, confirming that conditions defined and classified as HDUE do not have any association with serum total and HDL cholesterol levels, while this is the case for CHD. This is a fundamental, although partial, confirmation of similar findings reported by our research group using the same definitions, classification, and approach and dealing with national and international studies, in both genders, and during long and very long follow-up periods [4,5,6,7,8,9,10,11,12].

These facts suggest that only typical fatal CHD events are associated with serum cholesterol levels, and therefore with probable gross atherosclerotic lesions of coronary arteries, which likely should be not the case for HDUE. This concept was further stressed by dedicated analyses using the competing risks procedures, where it appeared that serum cholesterol was the critical and specific risk factor for CHD, while in the competing models for HDUE it showed a significant inverse relationship [16,17].

The specific advantage of the present analysis was the availability of a large denominator, still counterbalanced by a shorter follow-up period, but overall providing a substantial exposure in terms of person-years.

### 4.2. Unconfirmed Findings

We prudently stress the partial nature of this confirmation because other characteristics that were previously shown to be different between CHD and HDUE were not found (typically: age at death and multivariable coefficient for age), although this might be easily explained by a short follow-up period compared to other studies where we used data with a long follow-up, sometimes close to the extinction of the studied cohorts [4,5,6,7,8,9,10,11,12].

### 4.3. Limitations of the Study

Our study has limitations bound to several aspects. In particular, the baseline and the follow-up are confined to some years of the second half of last century. The number of risk factors used in the analysis were few and other factors were not available for this analysis, such as age at first occurrence of disease (that is the availability of incidence data), dietary patterns, and habitual physical activity, which was also proven in previous contributions to have a different relationship with CHD compared with HDUE [4,5,6,7,8,9,10,11,12]. The follow-up was short (around seven years), but somewhat compensated for by the size of the denominator. Moreover, measurements of inflammation markers were not considered in those times when the problem was not ripe for exploration in population studies.

### 4.4. General Considerations

We believe that when a heart manifestation (say, heart failure, atrial fibrillation, atrio-ventricular (AV) block, etc.) is not associated with an etiologically clearly defined heart disease (say, rheumatic, congenital, coronary, etc.) it should be, still arbitrarily, put in the “basket” of what we call HDUE, that—almost surely—is heterogeneous. The ICD codes do not always identify an “etiology”. For example, an atrial fibrillation alone, not accompanied by a diagnosis of rheumatic heart disease, CHD, or cardiomyopathy, etc., is, by definition, of “uncertain etiology” (or unknown etiology).

The fact is that this heterogenous “basket” is usually associated with most CVD risk factors, except serum cholesterol which seems to be the main “spy” of a true CHD involving the main coronary arteries. In this and other analyses, CHD has been used as counterpart of HDUE, because of these facts and because it is the most common heart disease. A confusion bound to ICD codes derives from the fact that “chronic ischemic heart disease” (in the absence of typical coronary syndromes) is not associated with serum cholesterol, and that in some studies heart diseases are classified as CHD, although not fully documented as CHD.

Coding of causes of death in this study was much less precise than in the majority of our previous analyses [3,4,5,6,8,9,10,11] where, beyond the availability of death certificates, in the majority of cases we could exploit information from periodic re-examinations, medical and hospital records, and interviews with family, hospital doctors, and relatives of the deceased and any other witness of the fatal events. Moving along this way we showed, in this and previous analyses on different populations, that the predicting role of serum cholesterol is the critical factor segregating CHD from HDUE.

### 4.5. Etiology of CHD and HDUE

In general, for CHD there is a large amount of evidence suggesting that lipid metabolism disorders, largely mediated by serum cholesterol, HDL cholesterol, and lipoproteins, with the help of other major—but nonspecific—risk factors, such as blood pressure, blood glucose, and some lifestyle behavior, represent the basic etiology of the disease that is preceded by the development of gross atheroma of major coronary arteries.

In the case of HDUE (or any term used to define it) a clear etiology is not known and apparently little efforts have been made to identify it, despite its frequency that in almost extinct male populations represents around 10% of total mortality and 20% of cardiovascular mortality [18].

Still, the most convincing evidence on the possible existence of CHD vs. HDUE cases can be found in very old pathology papers [19,20,21] showing that the size of myocardial scars had a bimodal distribution: large scars were strongly associated with gross atheroma and thrombosis, while this was not the case for small scars that, among other things, were not more common among cases with large scars. The hypothesis was made that infection, toxic, or allergic agents and other unknown causes could be responsible for cases with multiple small scars that led to diffuse sclerosis of the myocardium with muscle replacement and later to heart failure. Much later, the possible involvement of apoptosis in the origin of myocardial scars and sclerosis was raised [22], but it remains a possible pathogenetic step more than an etiological factor.

### 4.6. Risk Factors, HDUE, and Heart Failure

The literature is relatively poor of specific contributions because the terminology and concept of HDUE, suggested by our research group, are relatively recent and not spread enough.

A main issue is how heart failure is classified in the absence of typical coronary syndromes. For example, in the old Evans Country population study in the US, it was suggested that cases of heart failure in people aged 40 years or more, in the absence of reasonable causes, should be classified as CHD [23]. On the other hand, a position report of 1979 prepared by the International Society and Federation of Cardiology and the World Health Organization, stated that in cases of heart failure, in the absence of clear coronary syndromes, the diagnosis of CHD should remain only presumptive [24].

Another issue is how heart failure is considered in the various contributions of the literature. During the last 15 years much emphasis has been given to clinical studies where low serum cholesterol seemed associated with a poor prognosis in advanced stages of heart failure [25,26,27,28,29,30,31,32], but only two of them [27,32] were able to show that low serum cholesterol was associated with an improved outcome in patients with CHD, while it predicted a worse outcome in patients without CHD. It is clear that part of the above conclusions might be cofounded by the association of low serum cholesterol with malnutrition in older people with heart failure. Moreover, this is an entirely different problem compared with the prediction of events or mortality in subjects initially free from heart disease.

More recently, in population studies run in the US [33,34], Sweden [35], and Japan [36], levels of serum cholesterol were not (even inversely) correlated with the occurrence of heart failure, but only in the Japanese study [36] was a clear segregation of ischemic from nonischemic cases done showing the presence of an inverse association only in cases of nonischemic heart disease.

In a recent review of the Framingham Heart Study [37], the meaning of heart failure in the absence of a clear etiology was not considered with much detail, except showing its strong relationship with blood pressure levels (stronger than for CHD), while no mention was found for the age of occurrence or age at death.

Unfortunately, heart failure, which is an important component of what we call HDUE, is frequently considered a “disease” or a cause of death and not a clinical manifestation of etiologically defined (or undefined) heart conditions appearing in the course of their natural history.

In a large review on epidemiology of heart failure [38] dyslipidemia was classified as a minor risk factor of heart failure, together with “dietary factors”, smoking habits, and sedentary life style, but this statement was confounded by the list of “major” risk factors that included hypertension, left ventricular hypertrophy, myocardial infarction, valvular heart disease, obesity, and diabetes, some of which represent already established and etiologically defined heart diseases and not etiological or predictive factors of heart failure, by itself.

In another systematic worldwide review [39] of risk factors for heart failure, conditions such as rheumatic, valvular, and other etiologically defined heart diseases were also classified as risk factors of heart failure, but this does not help in identifying a possible etiology or at least risk factors of the underlying heart disease that causes heart failure when etiology is unclear or not defined like in the cases we call HDUE.

### 4.7. HDUE, Chronic CHD, and Heart Failure

In this material, a large proportion of deaths classified as HDUE were described as “chronic CHD”, which means that probably typical CHD syndromes were not apparent or known. This poses the problems of the assignment of causes of death, the vague definition of what “chronic CHD” may mean in the International Classification of Diseases and the suspicion that cases of HDUE have probably little to do with large coronary arteries atheroma.

“Chronic CHD”, is a rather vague terminology that, in the majority of cases, is accompanied by heart failure, but it is not clear why a heart failure should be attributed to CHD if typical CHD syndromes have not preceded or accompanied its occurrence.

Around the term and concept of heart failure, the literature is full of confusion since this syndrome is a rather common pathophysiological step during the natural history of most heart diseases in general. However, despite this, as mentioned above, it is frequently considered an independent disease and among its risk factors, several etiologically defined conditions are mentioned, such as rheumatic heart disease, nonrheumatic valvular diseases, congenital heart disease, etc., which represent a definite source of confusion.

### 4.8. Inflammation and CHD

Inflammation and its indicators have been associated with the initiation and progress of atheromatous lesions and as mediators and/or causes of acute CHD events [40,41,42]. However, in an early contribution, the addition of inflammation markers to a group of traditional risk factors compacted into the so-called “metabolic syndrome” did not add any valuable improvement of prediction [43].

Saturated fatty acids in the diet and smoking habits are among risk factors capable of triggering and sustaining inflammation. [44,45,46,47]. The role of proinflammatory components of the diet has been reported in a large study in women in the US [48]. However, some reports raised doubts about the causative role of inflammation, suggesting instead a prevalent role of mediator [49]. On the other hand, in a recent review, inflammation seems to play the role as an independent risk factor [50]. Moreover, in the CANTOS trial run in postmyocardial infarction patients, the occurrence of further events was reduced by the use of an innovative anti-inflammatory agent [50], which represents a new perspective in this area [51].

### 4.9. Inflammation and Heart Failure

During the last few years, a new development in the study of heart failure was represented by considering inflammation and its markers as a cause, mediator, or worsening agent of heart failure [52,53,54]. Reviewing some contributions, it appears that inflammation might be an important pathogenetic component of heart failure, but more likely an intermediate step rather than an etiologic agent. In fact, it seems that inflammation is more likely a mediator between external (infectious, toxic, allergic) or internal autoimmune [55] agents inducing and worsening heart failure. It is hard to accept that inflammation is an “etiologic agent” and not simply a reaction to external stimuli [56,57]. Moreover, it is difficult to find contributions segregating cases that we call HDUE from those characterized by a defined etiology (such as CHD, which are probably the majority), and rarely has this operation been clearly made.

The question we stress is the need to assess the role of inflammation in cases of heart failure without clear etiology and to identify its role as an etiologic agent and/or a mediator between known or unknown agents and the outcome. Among the risk factors of HDUE identified here, probably only smoking habits are causative of inflammation.

The recent interest for the relationship of inflammation with heart failure justifies the hypothesis made many years ago [19,20,21], where external agents, plus perhaps autoimmune processes [54], could be the etiologic factors, which are then enhanced by the inflammation contribution.

The possible role of fibroblasts in supporting heart failure is another important discovery [58].

## 5. Conclusions

Our findings tend to confirm that heart diseases manifested only as heart failure and arrhythmia—that we call HDUE—are not associated with serum cholesterol as a predictive risk factor and cannot be classified as atherosclerotic CHD, although their real etiology is still elusive.

The fact that HDUE is associated with other classical CHD risk factors, such as smoking habits, high blood pressure, and blood glucose, suggests the need to take care and intervene on these risk factors, irrespective of the future outcome as CHD or HDUE.

Since the disease manifests in most cases as heart failure, it seems mandatory to isolate cases without a clear etiology from other cases of heart failure representing the clear consequence of an etiologically defined heart disease, and to study its association with inflammation and other possible etiologic and pathogenetic agents.

## Figures and Tables

**Figure 1 medicina-55-00687-f001:**
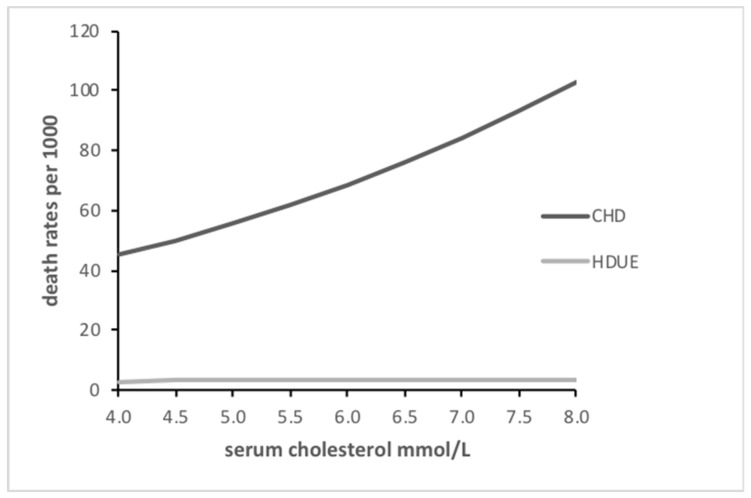
Relationship of total serum cholesterol with CHD and HDUE mortality.

**Figure 2 medicina-55-00687-f002:**
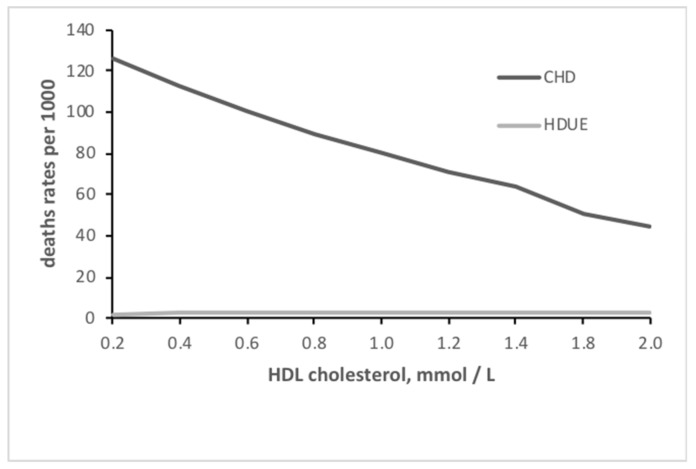
Relationship of high density lipoprotein (HDL) cholesterol with CHD and HDUE mortality.

**Table 1 medicina-55-00687-t001:** Median values (or proportions) of risk factors at entry examination in 47,167 cardiovascular diseases (CVD)-free subjects used in the analysis. *P* of difference was computed with test of proportions for categorial variables, *t*-test for continuous variables (based on means and standard deviations), and nonparametric Mann–Whitney test for continuous variables.

	MEN (N = 25,272)	WOMEN (N = 21,895)	*t*-Test or Test of Proportions	Mann-Withney Test	Both Sexes (N = 47,167)
	Median or Proportion and (*)	Median or Proportion and (*)	*p* of Difference Men Versus Women	*p* of Difference: Men vs. Women	Median or Proportion and (*)
Age, years	51 (14)	49 (15)	<0.0001	<0.0001	50 (14)
Never smokers, %	22.7 (0.3)	73.1 (0.3)	<0.0001	------	46.1 (0.2)
Ex-smokers, %	32.0 (0.3)	8.3 (0.2)	<0.0001	------	21.0 (0.2)
Current smokers, %	45.3 (0.3)	18.5 (0.3)	<0.0001	<0.0001	32.9 (0.2)
Body mass index, kg/m^2^	26.3 (4.2)	26.2 (5.9)	<0.0001	<0.0001	26.3 (4.9)
Systolic blood pressure, mmHg	135 (29)	131 (30)	<0.0001	<0.0001	133 (30)
Serum cholesterol, mmol/L	5.69 (1.53)	5.61 (1.58)	0.0162	<0.0001	5.66 (1.55)
HDL cholesterol, mml/L	1.24 (0.41)	1.46 (0.44)	<0.0001	<0.0001	1.32 (0.44)
Triglycerides, mg/dL	126 (87.7)	99 (66)	<0.0001	<0.0001	112 (80)
Plasma glucose, mg/dL	92 (16)	88 (15)	<0.0001	<0.0001	90 (15)
CHD, rate per 1000	7.7 (0.5)	1.2 (0.2)	<0.0001	------	4.7 (0.1)
HDUE, rate per 1000	4.9 (0.4)	1.1 (0.2)	<0.0001	------	3.2 (0.08)
All-causes, rate per 1000	50.0 (1.4)	18.0 (0.9)	<0.0001	------	34.9 (0.08)

(*) interquartile range for continuous variables; standard error for categorical variables.

**Table 2 medicina-55-00687-t002:** Hazards ratios of 11 risk factors derived from Cox proportional hazards models of coronary heart disease (CHD) and heart diseases of uncertain etiology (HDUE). Pooled sexes.

	Delta for Hazards Ratios	Hazards Ratio	95% CI	*p* of Coefficient
CHD				
Sex	1	4.21	2.66–6.68	<0.0001
Age	5 years	1.44	1.32–1.58	<0.0001
Never smokers	reference	-----	-----	-----
Ex-smokers	1	1.27	0.83–1.95	0.2709
Current smokers	1	2.03	1.36–3.01	0.0005
Body mass index	4 kg/m^2^	0.96	0.83–1.12	0.6400
Systolic blood pressure	20 mmHg	1.38	1.22–1.55	<0.0001
Serum cholesterol	1 mmol/L	1.24	1.11–1.39	0.0002
HDL cholesterol	0.35 mmol/L	0.81	0.69–0.95	0.0098
Triglycerides	90 mg/dL	0.96	0.86–1.08	0.5095
Plasma glucose	20 mg/dL	1.13	1.05–1.22	0.0010
**HDUE**				
Sex	1	2.99	1.80–4.96	<0.0001
Age	5 years	1.46	1.30–1.63	<0.0001
Never smokers	reference	-----	-----	-----
Ex-smokers	1	1.48	0.89–2.46	0.1271
Current smokers	1	1.92	1.19–3.08	0.0075
Body mass index	4 kg/m^2^	0.98	0.82–1.18	0.8658
Systolic blood pressure	20 mm Hg	1.28	1.11–1.49	0.0008
Serum cholesterol	1 mmol/L	1.03	0.89–1.19	0.7382
HDL cholesterol	0.35 mmol/L	1.01	0.84–1.21	0.9297
Triglycerides	90 mg/dL	0.99	0.85–1.16	0.9109
Plasma glucose	20 mg/dL	1.15	1.05–1.26	0.0021

**Table 3 medicina-55-00687-t003:** Hazards ratios of 11 risk factors derived from Cox proportional hazards models of CHD and HDUE. Men only.

	Delta for Hazards Ratios	Hazards Ratio	95% CI	*p* of Coefficient
CHD				
Age	5 years	1.42	1.29–1.57	<0.0001
Never smokers	reference	-----	-----	-----
Ex-smokers	1	1.37	0.85–2.22	0.2003
Current smokers	1	2.20	1.40–3.47	0.0006
Body mass index	4 kg/m^2^	1.05	0.89–1.25	0.5558
Systolic blood pressure	20 mmHg	1.38	1.21–1.56	<0.0001
Serum cholesterol	1 mmol/L	1.22	1.08–1.38	0.0012
HDL cholesterol	0.35 mmol/L	0.81	0.68–0.96	0.0198
Triglycerides	90 mg/dL	0.97	0.87–1.09	0.6269
Plasma glucose	20 mg/dL	1.16	1.00–1.19	0.0414
**HDUE**				
Age	5 years	1.43	1.26–1.62	<0.0001
Never smokers	reference	-----	-----	-----
Ex-smokers	1	1.22	0.70–2.13	0.4817
Current smokers	1	1.69	1.00–2.87	0.0513
Body mass index	4 kg/m^2^	1.00	0.81–1.24	0.9816
Systolic blood pressure	20 mm Hg	1.35	1.15–1.58	0.0002
Serum cholesterol	1 mmol/L	1.10	0.94–1.29	0.2477
HDL cholesterol	0.35 mmol/L	1.00	0.82–1.23	0.9790
Triglycerides	90 mg/dL	0.95	0.80–1.13	0.5710
Plasma glucose	20 mg/dL	1.16	1.06–1.28	0.0021

**Table 4 medicina-55-00687-t004:** Comparison of multivariable coefficients of Cox proportional hazards models of CHD and HDUE. Pooled sexes.

	CHD		HDUE		*p* of *t*-Test of Difference
	Coefficient	SE	Coefficient	SE	
Sex	1.4379	0.2353	1.0943	0.2587	0.3270
Age	0.0731	0.0093	0.0752	0.0114	0.8886
Never smokers	reference	----	----	----	----
Ex-smokers	0.2409	0.2188	0.3940	0.2582	0.6528
Current smokers	0.7063	0.2018	0.6500	0.2430	0.8572
Body mass index	−0.0091	0.0195	−0.0039	0.0233	0.8650
Systolic blood pressure	0.0159	0.0030	0.0125	0.0037	0.4778
Serum cholesterol	0.2136	0.0578	0.0248	0.0741	0.0444
HDL cholesterol	−0.5983	0.2315	0.0238	0.2692	0.0802
Triglycerides	−0.0004	0.0006	−0.0001	0.0009	0.8414
Plasma glucose	0.0063	0.0019	0.0071	0.0023	0.7918

SE: standard error.

**Table 5 medicina-55-00687-t005:** Differential role of risk factors and other characteristics between HDUE and CHD (present analysis and references [3,4,5,6,7,8,9,10,11]).

	HDUE	CHD
More prevalent in male	yes	yes
Higher role of age	yes	no
Direct relationship with smoking habits	yes	yes
Direct relationship with blood pressure	yes	yes
Direct relationship with body mass index	no	no
Direct relationship with serum cholesterol	no	yes
Inverse relationship with HDL cholesterol	no	yes
Direct relation shop with triglycerides	no	no
Direct relationship with plasma glucose	yes	yes
Protective effect of mediterranean diet	no	yes
Protective effect of vigorous physical activity	no	yes
Earlier age of occurrence	no	yes
Higher age at death	yes	no

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
