# Peer review of "Epidemiology of Heart Disease of Uncertain Etiology: A Population Study and Review of the Problem"

_medicina, 2019, doi:10.3390/medicina55100687_

Round 1

Reviewer 1 Report

This manuscript is presenting findings of two heart diseases groups, CHD and HDUE, have different association with predictive risk factors, specifically total serum cholesterol and HDL-cholesterol, while other predictive factors were not significantly different.

Minor revision:

1. why you didn't include more patients from later years (your data collected 30-35 y ago).

2. I can't find limitations section.

3. There are some grammar mistakes.

Author Response

REPLIES TO Referee 1

This manuscript is presenting findings of two heart diseases groups, CHD and HDUE, have different association with predictive risk factors, specifically total serum cholesterol and HDL-cholesterol, while other predictive factors were not significantly different.

Minor revision:

why you didn't include more patients from later years (your data collected 30-35 y ago). REPLY. In the Introduction the authors have shortly described analyses conducted on several different studies, that is all those available to them. All of them were started in the central part of last century, including the present one. I can't find limitations section. REPLY Some of the limitations of the study are scattered in the first 3 Sections of Discussion. Despite this, the authors have added another Section in Discussion (Section 4.3 in the new version) where limitations are given alltogether. Consequently, previous Section 4.2 has been shortened.

There are some grammar mistakes. REPLY. A language revision has been performed.

Reviewer 2 Report

Comments to the Author

Menotti et al. sought to investigate the difference in patient populations characterized under Heart Diseases of Uncertain Etiology (HDUE) and those classified as coronary heart disease (CHD). The authors hypothesized that CHD and HDUE are probably two different morbid conditions and addressed inflammation as a potential co-factor to differentiate between CHD and HDUE.

Major comments:

The paper is well designed and the results seem sound. The paper addresses an often overlooked clinical topic and is therefore of interest to readers, however, results are of rather descriptive nature as the entity of HDUE is somehow heterogeneous.

Are there data on inflammatory parameters available, eg. CRP, leucocytes, neutrophils? As the authors discuss inflammation as a major stimulus, this would be of great interest. If this is not possible, please add a paragraph on limitations. Please state that ROC curves only show data of the men subgroup in a legend.

Minor comments:

Please do a spell check, e.g. =< 0.005, some mark-ups

Author Response

REPLY to Referee 2

Comments to the Author

Menotti et al. sought to investigate the difference in patient populations characterized under Heart Diseases of Uncertain Etiology (HDUE) and those classified as coronary heart disease (CHD). The authors hypothesized that CHD and HDUE are probably two different morbid conditions and addressed inflammation as a potential co-factor to differentiate between CHD and HDUE.

 Major comments:

The paper is well designed and the results seem sound. The paper addresses an often verlooked clinical topic and is therefore of interest to readers, however, results are of rather descriptive nature as the entity of HDUE is somehow heterogeneous.

Are there data on inflammatory parameters available, eg. CRP, leucocytes, neutrophils?

As the authors discuss inflammation as a major stimulus, this would be of great interest. If this is not possible, please add a paragraph on limitations.

REPLY. As stated in the Introduction, inflammatory parameters were not measured since, those times, the problem was not fashion and some measurements not easy at population level.

Please state that ROC curves only show data of the men subgroup in a legend.

REPLY. This suggestion has been applied.

Minor comments:

Please do a spell check, e.g. =< 0.005, some mark-ups 

REPLY. This point has been corrected as p<0.05

Round 2

Reviewer 2 Report

The manuscript has improved and all questions were adressed.

This manuscript is a resubmission of an earlier submission. The following is a list of the peer review reports and author responses from that submission.

Round 1

Reviewer 1 Report

This is a large study that attempts to differentiate between coronary heart disease and heart disease of uncertain etiology.

It seems as if the 2 groups are already distinguished, as evidenced by their having different ICD codes.  I’m not sure there is a utility in grouping all heart related, non CHD problems in to the basket of HDUE.  HDUE implies uncertain etiology but the ICD codes say some of the etiology, such as arrhythmia.  I think it is intuitive that people with CHD and “HDUE” would have different risk factors, i.e. the it is known that risk factors for arrhythmias are different than for CHD. 

Also, in the manuscript there are a few terms that need to be defined, for example on Page 2 line 43 – the authors say this represents and improvement over code412, but do not explain what code 412 is.  They also refer to Euro Score and Rifle projects, but do not explain what those are

Author Response

Referee 1

English language and style

( ) Extensive editing of English language and style required 
(x) Moderate English changes required 
( ) English language and style are fine/minor spell check required 
( ) I don't feel qualified to judge about the English language and style 

Yes

Can be improved

Must be improved

Not applicable

Does the introduction provide sufficient background and include all   relevant references?

( )

(x)

( )

( )

Is the research design appropriate?

( )

( )

(x)

( )

Are the methods adequately described?

(x)

( )

( )

( )

Are the results clearly presented?

( )

( )

(x)

( )

Are the conclusions supported by the results?

( )

( )

(x)

( )

Comments and Suggestions for Authors

This is a large study that attempts to differentiate between coronary heart disease and heart disease of uncertain etiology.

QUESTION 1

It seems as if the 2 groups are already distinguished, as evidenced by their having different ICD codes. I’m not sure there is a utility in grouping all heart related, non CHD problems in to the basket of HDUE. HDUE implies uncertain etiology but the ICD codes say some of the etiology, such as arrhythmia. I think it is intuitive that people with CHD and “HDUE” would have different risk factors, i.e. the it is known that risk factors for arrhythmias are different than for CHD. 

ANSWER 1

The authors do not fully agree with the above comments. Their point is that when a heart manifestation (say, heart failure, atrial fibrillation, AV block etc) is not associated with an etiologically clearly defined heart disease (say, rheumatic, congenital, coronary, etc) it should be, still arbitrarily, put in the “basket” of HDUE, that- almost surely- is heterogeneous.

The ICD codes do not always identify an “etiology”. For example, an atrial fibrillation alone not accompanied by a diagnosis of rheumatic heart disease, of CHD, or cardiomyopathy, etc, is by definition of “uncertain etiology” (or unknown etiology).

The fact remains that this heterogenous “basket” is usually associated with most CVD risk factors, except serum cholesterol which seems the main “spy” of a true CHD involving the main coronary arteries. CHD, has been used as counterpart of HDUE, because of these facts and because it is the most common heart disease. A clear confusion of ICD codes derives from the fact that “chronic ischemic heart disease” (in the absence of typical coronary syndromes) is not associated with serum cholesterol. Moving along this way we showed, in this and previous analyses on different populations, that the predicting role of serum cholesterol is the critical factor segregating CHD from HDUE.

The above concepts have been partly presented in the revised text.

SEE Material and Methods (ICD coding) and new paragraph 4.3. in Discussion.

QUESTION 2

Also, in the manuscript there are a few terms that need to be defined, for example on Page 2 line 43 – the authors say this represents and improvement over code 412, but do not explain what code 412 is.

ANSWER 2

In the revised version a detailed explanation is given about code 412 and in addition also for code 414 of the ICD.

QUESTION 3

They also refer to Euro Score and Rifle projects, but do not explain what those are.

ANSWER 3

In the revision a short paragraph is added giving details about the EURO-SCORE and the RIFLE projects.

NOTE. The English language has been revised.

Reviewer 2 Report

In this manuscript, Menotti et al tried to investigate the characteristics and risk factor profiles in two different cardiac conditions --- coronary heart disease (CHD) vs Heart diseases of uncertain etiology (HDUE), using a large Italian cohort. The study is well done and provides important information and insights for both interventional and research purposes.

Recommendations for minor changes:

1)    Page 1, line 41, I suggest ICD-10 codes should also be mentioned and discussed since this is the ICD codes that are used in clinical practice currently

2)    This article also demonstrates a very important concepts that CHD and HDUE share many common risk factors (smoking, high glucose level, high blood pressure), and control these risks factors may decrease the mortality for both conditions. I would suggest authors emphasis this more in the discussion.

Author Response

Referee 2

English language and style

( ) Extensive editing of English language and style required 
( ) Moderate English changes required 
(x) English language and style are fine/minor spell check required 
( ) I don't feel qualified to judge about the English language and style 

Yes

Can be improved

Must be improved

Not applicable

Does the   introduction provide sufficient background and include all relevant   references?

(x)

( )

( )

( )

Is the   research design appropriate?

(x)

( )

( )

( )

Are the methods   adequately described?

(x)

( )

( )

( )

Are the   results clearly presented?

(x)

( )

( )

( )

Are the   conclusions supported by the results?

(x)

( )

( )

( )

Comments and Suggestions for Authors

In this manuscript, Menotti et al tried to investigate the characteristics and risk factor profiles in two different cardiac conditions --- coronary heart disease (CHD) vs Heart diseases of uncertain etiology (HDUE), using a large Italian cohort. The study is well done and provides important information and insights for both interventional and research purposes.

Recommendations for minor changes:

QUESTION 1

Page 1, line 41, I suggest ICD-10 codes should also be mentioned and discussed since this is the ICD codes that are used in clinical practice currently.

ANSWER 1

In the revised version a paragraph has been added to discuss the use of the WHO ICD-10 (including a Reference).

SEE Introduction.

QUESTION 2

This article also demonstrates a very important concepts that CHD and HDUE share many common risk factors (smoking, high glucose level, high blood pressure), and control these risks factors may decrease the mortality for both conditions. I would suggest authors emphasis this more in the discussion.

ANSWER 2

A sentence on this issue has been added in the Discussion (Conclusions).

NOTE. The English language has been revised.